# Reference Range of Kaolin-Activated Thromboelastography (TEG) Values in Healthy Pet Rabbits (*Oryctolagus cuniculus*)

**DOI:** 10.3390/ani13142389

**Published:** 2023-07-23

**Authors:** Tiziana Bassan, Josep Pastor, Beatriz Agulla, Oriol Jornet, Jaume Martorell

**Affiliations:** 1Fundació Hospital Clínic Veterinari, Facultat de Veterinària, Universitat Autònoma de Barcelona, 08193 Cerdanyola del Vallès, Spain; tiziana.bassan@gmail.com; 2Departament de Medicina i Cirurgia Animals, Facultat de Veterinària, Universitat Autònoma de Barcelona, 08193 Cerdanyola del Vallès, Spain; josep.pastor@uab.cat (J.P.); beatriz.agulla@uab.cat (B.A.); oriol.j.r.89@gmail.com (O.J.)

**Keywords:** haemostatic disorder, rabbit, reference range, thromboelastography, viscoelastic method

## Abstract

**Simple Summary:**

Coagulation is a protective mechanism that prevents blood loss. The prothrombin time and activated partial thromboplastin clotting time are the most used laboratory tests used to assess coagulation status in all mammals. However, these tests do not accurately reflect the in vivo coagulation process due to complex interactions between blood cells and coagulation elements. New cell-based models incorporate the role of cells to explain the formation/destruction of the blood clot through viscoelastic laboratory techniques, such as thromboelastography (TEG). Thromboelastography reference ranges have been determined for humans, dogs, cats, horses, rats, mice, and laboratory rabbits but they have not been previously defined in pet rabbits. The aim of the article is to establish the reference-range values for TEG parameters in pet rabbits. Included in this study were 24 healthy rescue pet rabbits. A TEG 5000 Thromboelastograph Hemostasis System with citrated kaolin-activated blood was used. The reference ranges obtained for reaction time (R) were 1.4–6.9 min; clot formation time (K) was 0.8–2.2 min; α-angle was 65.8–82.2 degrees; maximal amplitude (MA) was 53.7–73.5 mm; measure of clot strength/firmness (G-value) was 5796.6–13,885.9 dyn/cm^2^; the percentage of clot lysis in 30 min (LY30%) was 0–41.5%. This study provides the reference ranges of TEG in pet rabbits.

**Abstract:**

Thromboelastography (TEG) is a viscoelastic technique that allows the examination of both cellular and plasma protein clotting factors. Thromboelastography helps to investigate the underlying coagulopathy and to monitor therapeutic modalities. Although viscoelastic techniques have been used in human and veterinary medicine, reference ranges in pet rabbits are missing. The objective of this study is to establish the reference-range values of TEG parameters in healthy pet rabbits. 24 healthy pet rabbits of different breeds were included: 16 crossbreeds, four Californians, two lops, one lionhead, and one angora. Four rabbits were less than one year old and 20 were older than one year. Twelve rabbits were neutered females, 10 neutered males, and two were intact females. Health status was assessed through a physical examination, a complete blood work, and a coagulation profile. A TEG 5000 Thromboelastograph Hemostasis System was used with kaolin-activated citrated whole blood. All samples were analysed 30 min postextraction. The TEG reference ranges were reaction time (R) 1.4–6.9 min; clot formation time (K) 0.8–2.2 min; α angle 65.8–82.2 degrees; maximal amplitude (MA) 53.7–73.5 mm; measure of clot strength/firmness (G-value) 5796.6–13,885.9 dyn/cm^2^; and percentage of clot lysis in 30 min (LY30%) 0–41.5%. This study provides the reference ranges of TEG in pet rabbits.

## 1. Introduction

Haemostasis is a vital protective mechanism that prevents blood loss by sealing sites of injury in the vascular system [1]. It depends on a balance between procoagulant and anticoagulant processes and on fibrinolysis which is responsible for clot dissolution [2]. Traditionally, plasma protein has been used to study the haemostatic process in vivo to assess the extrinsic, intrinsic, and common pathways of the coagulation process [3]. The new cell-based model incorporates the role of cells in the coagulation process [2,3]. The cell-based model describes haemostasis as occurring in three overlapping phases: initiation, amplification, and propagation, which occur on the surface of cellular components of blood [4]. Whole-blood-based assays, such as thromboelastography (TEG), permit the evaluation of in vivo haemostasis more realistically by taking into consideration all the intravascular elements [5]. The systems most used to assess viscoelastic blood properties are thromboelastography (TEG) and rotational thromboelastometry (ROTEM). Both systems describe the movement of the pin during clot formation [5]. In TEG, the cup oscillates around the pin [5]. The cup movement is monitored by the pin hanging on a torsion wire. Therefore, the pin movement is converted into an electrical signal and registered by TEG software [6]. By contrast, in ROTEM, the cup remains immobile with an oscillating pin. TEG and ROTEM operated in similar ways; however, the results obtained are not comparable [7]. Three sample types can be used to perform a TEG: whole native blood, citrated whole blood activated with human recombinant tissue factor, or citrated whole blood activated with kaolin [5]. The human recombinant tissue factor is not commercially available, so the use of kaolin is preferred [5]. Thromboelastography was implied to monitor the coagulation process during surgery in human medicine and for animal experiments [5]. Now, this technique is used in veterinary medicine to diagnose and treat hypocoagulable and hypercoagulable status and hyperfibrinolysis activation, inflammation, infection, or neoplastic diseases [5,8,9,10]. In the exotic animal clinical field, viscoelastic techniques for animals are scarce. The rabbit has been used for a research model [11,12,13]. Thromboelastography reference ranges have been described for dogs [14,15,16], cats [17], horses [18], rats [19], pigs [5,20,21], and mice [19]. Reference ranges have until now been missing in pet rabbits for routine use in veterinary practice to assess coagulation status, although previous studies have been performed in laboratory rabbits. One study reported TEG values with human recombinant tissue factor and Cytochalasin D for 12 laboratory rabbits [22]. Another human medicine research study was investigating the clinical usefulness of plasminogen activator inhibitors as a haemostatic agent in human medicine, comparing the kaolin-activated TEG values with a control group of three laboratory rabbits [20]. A study with 12 New Zealand rabbits demonstrated a coagulation reduction in cases of hypothermia with celite-activated TEG [11]. TEG was used to study the coagulative status of different abdominal injuries in rabbits [23]. Another study demonstrated that thromboelastography performed with the recalcified blood of 11 New Zealand White Rabbits is more sensitive to detecting changes in heparin activity than aPTT and ACT [24]. There are also some studies about implying rabbits as animal models to study envenomation and how coagulation status varies with haemodilution with different colloids solutions [25,26].

Thromboelastography values are represented with a graphic curve. The “*x*-axis” represents the time, while the “*y*-axis” represents the amplitude of the clot in millimetres (mm). The reaction time (R) is measured in minutes. It represents the time the clot takes to reach an amplitude of two mm and where fibrin polymers are first produced. Clot formation time (K) is also measured in minutes and it represents the time the clot takes to reach an amplitude of 20 mm. The α angle is measured in degrees and is the angle formed by the tangent to the curve at K time and *y*-axis; it represents the acceleration of clot formation and the kinetics of fibrin formation and cross-linkage. Maximal amplitude (MA) is the maximal amplitude reached by the clot, representing clot strength, and it depends on platelet number and fibrinogen concentration. Clot lysis (LY30%) is measured in minutes and it is the % of clot lysis at 30 min from the clot formation. The G-value is the shear elastic module and is measured in Dynes/cm^2^ and it is derived from the formula G = 5000 × MA/(100 − MA) [27,28].

The aim of this study is to establish reference ranges of TEG values in pet rabbits.

## 2. Materials and Methods

### 2.1. Animals Selected

Twenty-four healthy rescued pet rabbits were recruited as part of routine check-ups performed at the Veterinary Teaching Hospital, Department of Veterinary Clinical Sciences, Faculty of Veterinary Medicine, Autonomous University of Barcelona, Spain. Owner consent forms have been obtained for all subjects included in the study. Ages were estimated to be between six months and four years because of the animals’ rescue status. Sex was distributed between 12 neutered females, 10 neutered males, and two intact females. The rabbits were of mixed breeds, distributed as follows: 16 crossbreeds, four Californians, two lops, one lionhead, and one angora. All the rabbits were adults: four young adults (between six months and one year) and 20 adults (over one year). They were fed a commercial diet made of pellets for rabbits (brands: Oxbow, Versele Laga and Beaphar) ad libitum and had free access to hay and water. They were vaccinated against myxomatosis and rabbit haemorrhagic disease strains one and two and treated against ectoparasites and endoparasites annually. They showed no signs of illness in the two weeks preceding or two weeks following sample collection and received no treatment before screening. Signalment was recorded for each individual rabbit. Their weight ranged between 1.2 to 4.0 kg and their body condition score was assessed with a Rabbit Size-O-Meter (available from: https://www.pfma.org.uk/rabbit-size-o-meter, accessed on 1 July 2022). The inclusion criteria for the study were: healthy according to their medical history (no bleeding tendencies), unremarkable physical examination (body condition score assessment, thoracic auscultation, abdominal palpation, and dental examination with an otoscope), and baseline blood laboratory diagnostic evaluation (complete blood count (CBC), plasma biochemical profile, and coagulation profile) within normal reference-range values. No external funding has been received and the study has been approved by the institutional review board.

### 2.2. Sample Collection and Processing

No animals were fasted, sedated, or anaesthetized prior to blood collection. Rabbits were gently manually restrained. Blood samples were collected by left jugular venepuncture with a 23 Gauge needle and a five ml plastic syringe. Samples were immediately transferred to different tubes in the following order: one ml of blood was transferred into two 0.5 mL citrate tubes (ratio citrate:blood 1:9, Aquisel S.L.U., Abrera, Spain), one for coagulation tests and one for TEG profiles; 0.5 mL of blood was stored in K3EDTA tubes (K3E/EDTA 3K, Aquisel S.L.U., Abrera, Spain) for haematology analysis and 0.5 mL placed in a heparin tube (MiniCollect^®^ LH Lithium Heparin Blood Collection Tube, Greiner Bio-One, Abrera, Spain Kremsmünster, Austria) for biochemistry. Samples were gently moved to ensure a good distribution of the anticoagulant stored at room temperature and processed a maximum of 30 min later. Any tube with visible signs of clotting was excluded from the process. The blood in the heparin tubes and one 0.5 mL citrate tube was centrifuged at 3500 rpm for five minutes. One aliquot of 0.5 mL of citrated blood was stored at −20 °C within 30 min of serum harvesting to determine prothrombin time (PT), partial thromboplastin time (PTT), and fibrinogen. For haematology, “SYSMEX XN−1000” with the software “XN-V Series ver. 3.07-00” was used. For biochemistry, “Beckman Coulter AU480 Analyser” with software “Beckman AU480 ver. V.1.A1” was used. All the reagents are OSR (Olympus System Reagent^®^, Beckman Coulter^®^, O’Callaghan’s Mills, Ireland). The coagulation profile consisted of activated prothrombin time PTTa, prothrombin time PT, and fibrinogen concentration. ST art “DIAGNOSTICA STAGO” was used (Asnieres-sur-seine, France). All analyses were performed following the manufacturers’ recommendations and with quality controls for assay conditions and instrumentation.

### 2.3. Thromboelastography

The thromboelastographic technique was performed with a computerized thromboelastograph “TEG 5000 Hemostasis Analyzer©” (Haemonetics©, Ferrer Farma, Barcelona, Spain) with software “TEG 4.2.3”. A sample of 0.5 mL of blood kept in sodium citrate was used to perform the procedure. Subsequently, 500 µL of blood was transferred to an Eppendorf tube, with 20 µL of kaolin (Haemonetics S.A., Signy-Avenex, Switzerland), and gently inverted five times. Finally, 340 µL of citrated kaolin blood was pipetted into the cup where 20 µL calcium chloride (CaCL_2_) had been previously added to start the clotting process. Analysis was performed following a standardised 30 min delay. The procedure normally lasts 120 min. The quality control was settled daily with levels I and II (Haemoetics©, Ferrer Farma, Barcelona, Spain).

### 2.4. Statistical Analysis

A Shapiro–Wilk test was used to test for the normality distribution of the studied parameters. Statistical significance was set at *p* < 0.05. When two independent variables were compared, a Student’s T test for parametric or U of Mann–Whitney for nonparametric values was carried out. When three or more variables were compared, an Anova parametric test or Kruskal–Wallis test for nonparametric values was used. Statistical significance was set at *p* < 0.05. An informatics software, Graph Pad Prism 8 (GraphPad Software, San Diego, CA, USA), was used for normality and comparison studies. Reference ranges were determined using a Microsoft Excel spreadsheet.

## 3. Results

Haematological, biochemical, and coagulation are summarised in Table 1, Table 2 and Table 3.

An illustration of the TEG 5000 viscoelastic tracing of citrated whole blood and kaolin activation is shown in Figure 1. Variables R, K, Angle, and Lys30 were not normally distributed, while MA and G were normally distributed. No statistically significant differences for age, sex, or breed were detected (*p* > 0.05) among the animals studied; thus, the results of the TEG parameters were represented as a single population.

The TEG results are reported in Table 4: reaction time R = 1.4–6.9 min; clot formation time K = 0.8–2.2 min; α angle = 65.8–82.2 degrees; maximal amplitude MA = 53.7–73.5 mm; G-value = 5796.6–13,885.9 dyn/cm^2^; and % of clot lysis in 30 min LY30% = 0–41.5%. The histograms demonstrating the distribution of TEG parameters are visualized in Figure 2.

## 4. Discussion

The viscoelastic techniques are influenced by several factors; therefore, standardization is needed. Preanalytical variables, such as sample collection, handling, transport, storage time, and temperature activators can affect the results. It is recommended that each laboratory set up quality-control protocols according to guidelines to standardize sample acquisition, handling, and assay protocol [7,28,29]. Each centre should create its own reference intervals [28].

For blood collection, a jugular venepuncture was recommended in dogs, cats, and horses [7]. In the current study, all the rabbit blood samples were obtained by jugular venepuncture to avoid traumatic blood collection as much as possible. The use of 21-G needles or larger needles has been suggested for taking blood samples to ease free-flowing blood [30]. However, considering the challenges of blood collection in conscious rabbits and the small vessel diameter, a 23-G needle was used.

Prolonged or short blood-storage time can affect TEG results [30]. In dogs, canine-citrated whole blood for TEG analysis with human recombinant TF was stored at 30 and 120 min after sample collection [30]. Long blood-storage time results in a tendency to hypercoagulability, compared to the results at 30 min [30]. Another study of TF-activated TEG in horses considered 30-, 60- and 120-min storage time and it showed the same hypercoagulable results in the case of longer rested samples [31]. Considering this factor, a systematic evaluation of evidence on veterinary viscoelastic testing recommended that blood samples should be rested for 30 min prior to blood analysis [30]. The same term and conditions were fixed at 30 min in the current study. More research into storage time is needed in rabbits. 

The use of an activator standardizes the assay result, establishing the moment of clot initiation and helping the stabilisation of clot formation [29]. In humans, it has been demonstrated that kaolin-activated TEG makes assays more stable than the ones without an activator [29,32]; also, a correlation between non-kaolin-activated and kaolin-activated, for both citrated and native blood [33]. In veterinary medicine, TEG was performed in 40 healthy dogs to see if there was a correlation between samples with or without kaolin activation. The TEG results mainly agree between kaolin-activated and native samples but, when citrated blood was activated with kaolin, the clot formation time was shortened and amplitude was increased, showing a tracing that indicated implied hypercoagulability [34]. In cats, it was seen that results from three different TEG assays (without an activator, with human recombinant tissue factor or kaolin) cannot be used interchangeably [17]. This evidence underlines the fact that it is fundamental to consider the activator employed in the technique. The kaolin activator was used in the current study.

Differences between age, sex, or breed have been described in human and veterinary medicine. In human medicine, no age-related differences were observed in a study of kaolin-activated TEG values between children and adults [35]. Nevertheless, a TEG study with citrated native blood samples showed that reaction times and G-values were significantly lower in neonates than in adults [6]. In the current study, only adult animals were selected (between six months and one year and more than one year) and no differences in the TEG values were detected. Regarding sex, no differences in kaolin-activated TEG values were detected in dogs [16]. In the present study, no differences in rabbit TEG values were detected for sex, as the majority of animals were neutered. Reference ranges for kaolin-activated TEG in dogs were established on a population mainly represented by beagles and German shepherds, so the study might not be representative of the canine population [16]. In the current study, most of the rabbits selected were crossbreed species, which is probably the most frequent animal seen in clinical practice, but not representative of every breed.

In exotic animal medicine, it is difficult to obtain many healthy animals to calculate the reference-range values for a species. According to the American Society for Veterinary Clinical Pathology (ASVCP) reference interval guidelines, in the case of ≥20 and <40 individuals, parametric or robust methods are indicated to calculate the 90% confidence interval [36]. In our study, only 24 animals were available, and the reference range was described using robust methods.

Accordingly, to the previous literature, in a study, thromboelastographic parameters with tissue factor of 12 rabbits were reported as mean ± standard deviation: reaction time R = 2.5 ± 0.6 min; clot formation time K = 1.0 ± 0.2 min; α angle = 78.8 ± 2.7 degrees; maximal amplitude MA = 76.3 ± 4.0 mm; and G-value = 16.556 ± 3314 dyn/cm^2^ [22]. Another human-medicine research study used a control group of three rabbits reporting the following kaolin-activated TEG parameters with mean and standard deviation: reaction time R = 4.0 ± 0.2 min; clot formation time K = 1.4 ± 0.1 min; and % of clot lysis in 30 min LY30% = 0.0 ± 0.0; MAG-value = 5.0 ± 0.3 dyn/cm^2^ [20].

In the current study, Lys30 was reported out of range in three animals. Prolongation of Lys30 suggests an enhanced fibrinolytic state or increased activation of the body’s clot-dissolving system. There are several possible reasons for prolonged Ly30%, such as disseminated intravascular coagulation (DIC), liver disease, certain medications like fibrinolytic, or hypercoagulative states, as well as technical factors or individual variations [28]. Technical factors described with prolonged Lys30 are poor sample handling, incorrect preparation of the reagents, or malfunctioning equipment. Of the animals out of range reported in our study, Lys30 was the only parameter out of the reference range. Considering that it is important to interpret the results in conjunction with a thorough clinical evaluation and clinical medical history, we hypothesize that the observed variation may reflect individual changes and, less probably, inherited disease of the animals or a technical problem.

In other species, the previous TEG parameters reported are the following. In humans, reference intervals for kaolin-activated thromboelastography are described as R = 5.3–9.3 min, K = 1.4–3.5 min, angle a = 48.8–72.2 degrees, and MA = 55.3–69.3 mm [35]. In healthy dogs, reference ranges for kaolin-activated thromboelastography were as follows (*n* = 56): R = 1.8–8.6 min; K = 1.3–5.7 min; angle a = 36.9–74.6 degrees; MA = 42.9–67.9 mm, and G = 3.2–9.6 Kdyn/cm^2^ [16]. In cats, thromboelastograph measurements with a kaolin range of citrated whole blood were R = 2.4–9.5 min, K = 1.2–3.9 min, alfa-angle = 45.5–73.5, MA = 46.8–66.1 mm, and Ly30 = 0.0–9.0 [17].

Some studies analysed TEG parameters in rabbit models. In one study, snake venom in contact with CO has been seen to be partially or completely inhibited in whole blood in vivo in eight New Zealand white rabbits [25], with similar results observed in sedated animals [37]. In the study of Nielsen of 2918 rabbits, the quantity of venom was seen to be 15 times more in vivo than in vitro; so, that confirms that it is necessary to study in vivo, as this study was performed to give a reference range for in vivo healthy pet rabbits.

Several studies studied through TEG how haemodilution can affect coagulation in laboratory rabbits. As previously seen, decreased haemostasis has been seen in plasma celite-activated TEG with the administration of Volumen and Pentalyte at different times, or with Ringer’s solution [38,39]. A possible application of reference-range parameters published is to study haemodilution in pet rabbits.

This is the first study to describe citrated kaolin-activated TEG values in pet rabbits in the veterinary literature.

Further investigation with a larger number of animals is recommended, considering different breeds for a better representation of the rabbit population. It is recommended to study rabbits with coagulation disorders since no pet rabbits with bleeding/thrombotic disorder have been analysed in this study.

The limitations of the current study were the reduced number of rabbits selected. Further analysis should have been employed to assess health status, such as urinalysis, an extended coagulation profile with thrombin time (TT), antithrombin (AT), fibrin D-dimer, and factor VIII activity. Regarding the TEG, further studies are needed to assess the use of different activators at different storage times for the blood sample.

The current study provides a reference range for citrated kaolin-activated TEG in pet rabbits.

## 5. Conclusions

This study provides a reference range for citrated kaolin-activated TEG in healthy pet rabbits. It is recommended that each laboratory standardize the procedure design, considering the analyser and type of activator to establish reference values for the correct interpretation of the results.

## Figures and Tables

**Figure 1 animals-13-02389-f001:**
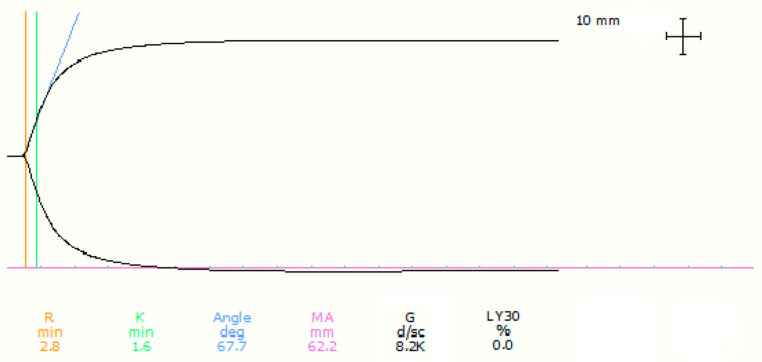
Example of the TEG graphical representation and analysis for a pet rabbit. R = reaction time, measured at the level of the orange line; K = clot formation time, measured at the level of the green line; Angle deg = α angle, measured at the level of the blue line; MA = maximum amplitude measured at the level of the violet line; G = measure of clot strength/firmness G-value; LY30% = percentage of clot lysis in 30 min.

**Figure 2 animals-13-02389-f002:**
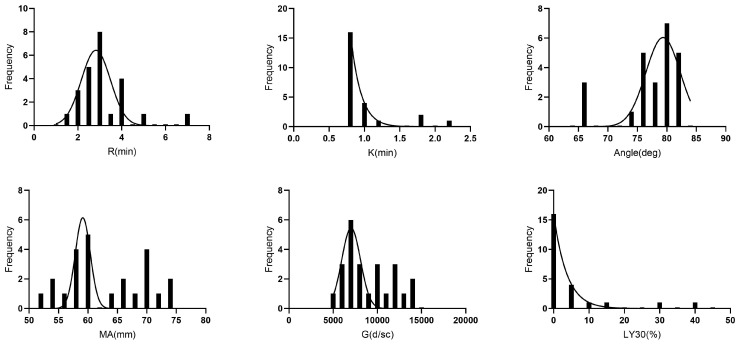
Histograms demonstrating the distribution of TEG data from the 24 healthy rescued pet rabbits. The dotted curve indicates the Gaussian curve.

**Table 1 animals-13-02389-t001:** Haematology parameters of *n* = 24 rabbits (mean, standard deviation, laboratory range).

Parameter (Unit)	Mean	Standard Deviation	Range
Haematocrit (%)	34	2.7	31.3–43.3
Erythrocytes (×10^6^/µL)	5.4	0.5	4.5–6.9
Haemoglobin (g/dL)	11.4	1	11.0–14.4
Mean corpuscular haemoglobin (Pg/cell)	21.2	1.2	19.4–23.8
Mean corpuscular haemoglobin concentration (g/dL)	33.6	0.8	32.3–34.5
Mean corpuscular volume (g/µm^3^)	63	2.6	59.0–70.1
Platelets (×10^3^/µL)	484.6	122.1	134–567
White blood cells (×10^3^/µL)	7.5	2.7	4.1–10.8
Neutrophils (×10^3^/µL)	2.8	1.3	1.1–7.4
Lymphocytes (×10^3^/µL)	4.1	1.6	0.5–6.5
Monocytes (×10^3^/µL)	0.2	0.1	0–3.7
Eosinophils (×10^3^/µL)	0.01	0.01	0–0.03
Basophils (×10^3^/µL)	0.4	0.3	0–0.4

**Table 2 animals-13-02389-t002:** Biochemistry parameters of *n* = 24 rabbits (mean, standard deviation, laboratory range).

Parameter (Unit)	Mean	Standard Deviation	Range
Alanine transferase (U/L)	75.6	25.5	52–80
Bilirubin (total) (mg/dL)	0.14	0.03	0.1–0.5
Chloride (mmol/L)	106	2.8	96–109
Cholesterol (mg/dL)	35.2	10.9	6–65
Creatinine (mg/dL)	1.1	0.2	1.0–2.2
Glucose (mg/dL)	150.5	15.7	109–161
Phosphorous (mg/dL)	3.4	0.9	3.0–6.2
Potassium (mmol/L)	4.5	0.4	3.4–5.1
Sodium (mmol/L)	142.8	2.4	138–148
Total protein (g/dL)	6.3	0.4	6.1–7.7

**Table 3 animals-13-02389-t003:** Fibrinogen, PT, PTTa of *n* = 24 rabbits (mean, standard deviation, laboratory range).

Parameter (Unit)	Mean	Standard Deviation	Range
Fibrinogen (mg/dL)	348	138.8	289.0–393.3
PT (s)	8.7	0.5	4.5–10.5
PTTa (s)	16.3	2.3	15.7–42.7

**Table 4 animals-13-02389-t004:** Thromboelastography values in healthy pet rabbits with *n* = 24. R = reaction time; K = coagulation time; α = angle α; MA = maximum amplitude; G = clot rigidity. RUD: robust untransformed data, SUD: standard untransformed data. Out of RI: out of reference interval.

Variable (Unit)	Distribution of Data	Mean	Standard Deviation	Median Range (Min–Max)	Lower 5% Reference Limit	Upper 95% Reference Limit	Method	Out of RI *n*/tot (%)
R (min)	Not normal	3.6	1.2	3 (1.4–6.9)	1.9	5	RUD	1/24 (4.2)
K (min)	Not normal	1	0.4	0.8 (0.8–2.2)	0.9	1.8	RUD	3/24 (12.5)
α Angle (degrees)	Not normal	77.4	5	79.0 (65.8–82.2)	65.9	82.1	RUD	3/24 (12.5)
MA (mm)	Normal	62.9	6.7	60.6 (52.4–73.54)	53.7	73.5	SUD	0/24
G (dyn/cm^2^)	Normal	8969.1	2711.7	7688.6 (5497.0–14,213.4)	5796.6	13,885.9	SUD	0/24
LY30 (%)	Not normal	4.7	10.1	0 (0–41.5)	0	25.7	RUD	3/24 (12.5)

## Data Availability

The datasets analysed in the present study are available from the corresponding author upon reasonable request.

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
