# Peer review of "Reference Range of Kaolin-Activated Thromboelastography (TEG) Values in Healthy Pet Rabbits (Oryctolagus cuniculus)"

_animals, 2023, doi:10.3390/ani13142389_

Round 1

Reviewer 1 Report

Interesting paper on a subject not treated before in rabbits even if availability is low. 

Remarks:

l97: you talk about "due to their rescue status" but you did not mention at the beginning of the paragraph that these rabbits are all rescue rabbits. It should be said there.

l96: I am bothered by the 4 years limit. Being rescue rabbits, honestly, I don't see at all how you can say that these rabbits have between 1 and 4 years. I have a rabbit at home who is 6 and I defy you to say that he is older than 4 unless told otherwise so I dougt you can be sure that these rabbits are not over 4. I think this issue has to be adressed and said differently.

L101: "commercial diet" ? Please be more precise: pellets ? Mixed seeds ? What amount ? ad libitum ?

l111: unclear. You mean "within" normal range ?? "With normal range" does not mean anything to me.

L111: perhaps these values should be somewhere ? A table ? See with editor is deemed necessary

L37 v. L137: unclear: at the beginning you say every sample was analysed 30mn post extraction and then you yalk of a dealy within 2 hours of collection ?

L.215: unclear. I don't understand the sentence precisely.

L.219: I d'ont agree. Mixed bredds represent most of the rabbits we see so it is probably representative of the general population of rabbits but not of every breed.

Reviewer 2 Report

The manuscript communicates reference ranges for thromboelastograpic parameters in healthy pet rabbits. The research gap is that such reference ranges have not been established, and the study aims to fill this gap. The methods and materials employed are relevant to the research objective.

There are, however, some points that the authors kindly needs to consider/amend:

1.  Introduction: It is stated that only 2 studies have previously used TEG analysis in rabbits. This is incorrect - e.g. a quick search on Pubmed using the search phrase TEG AND rabbit yields more than 2 studies.

2. Introduction: A supporting argument on the clinical relevance of performing TEG analysis in rabbits, and hence a further support of the rationale for doing the study, other than reference ranges are lacking will strengthen the manuscript, especially since the focus is on pet rabbits. In fact, one of the studies retrieved above concerned influence of hypothermia on TEG parameters in rabbits.

3. Material and Methods: - 2.2. Sample collection and processing: Please rephrase the section. In its present form, serum is used for coagulation assays (PT, PTT, fibrinogen) - and it may be misunderstood if citrated blood was centrifuged before TEG analysis. Further, the brand and manufacturer of the citrate tube must be noted as it is for the other tubes. Please, also note the tube size (was it tubes designed for 0.5 mL or 1mL ?)

4. Material and Methods: 2.3. Thromboelastography. 500uL blood and 20 uL kaolin is used. I believe that in most other cases, 1 mL blood is used. In the study referred to in the Introduction (ref 15) 1 mL of blood and 20 uL of kaolin was employed. - and in other studies (admitted on other species) specific silicated kaolin containing viles. What was the reason for using 500uL and 20 uL of kaolin in this study?

5. Material and Methods: 2.4 Statistical analysis. As the objective of the study is to establish reference ranges, the statistical procedure has to be referred and explained in this section. Also, there are a number of available software programs for calculating reference ranges e.g. Reference Value Advisor, which can be found at http://www.biostat.envt.fr/reference-value-advisor/. A short description of the difference between robust untransformed data and robust transformed data will increase readability and understanding of Table 1. 

6. Results: Please, include a table that communicates the summary statistics of at least WBC, RBC, PCV, HGB, Platelets, PT, PTT, fibrinogen

7. Results:  For small samples, it is recommended that all values be reported graphically in a dot plot or histogram and that estimates of the reference limits be compared using different methods, see e.g. Geffré et al. https://doi.org/10.1111/j.1939-165X.2009.00155.x

8. Results: Were any blood tudes excluded (ref. section 2.2 sample collection and processing)? Were hemolysis detected in some samples (see e.g. Bauer et al. https://doi.org/10.1111/j.1939-165X.2010.00224.x)

9. Discussion: Please review the included references as some, e.g. ref 18, is not the original reference but a citing reference for the original study.

10. Discussion: A section where comparison to the values found in other studies where TEG has been applied to rabbits should be included.

11. Table 1: the mean value for R is wrong (it should read 3.15)

Reviewer 3 Report

This is a nicely written manuscript and generally suitable for Animals. The study describes how reference ranges are established for kaolin activated thromboelastography values of healthy pet rabbits (Oryctolagus cuniculus). The study is based on citrated whole blood samples from 24 pet rabbits. The authors do not overinterpret the results but discuss the challenges. The manuscript could benefit from a clarification and some adjustments to the text to improve clarity and readability. 

No external funding has been received and the study has been approved the Institutional review board. Owner consent forms have been obtained for all subjects included in the study. 

Below comments:

Simple summary:

Line 14: it is stated ptt and pt are the most used laboratory tests. But it is not stated in which species? Is it in rabbits, companion animals or all mammals?

Line 15: “Other coagulation element” is stated, what does “other” reflect? It could eventually be deleted? or explained.

Line 17. The sentence is beginning with the abbreviation TEG. After a “.” an abbreviation should be spelled out. This is also the case in the rest of the manuscript as well. e.g. also in line 28 and 78.

Line 18: Normal values have been established in research rabbits, they should be considered to be mentioned as well.

Abstract:

Numbers under 10 is most often spelled with letters, so it is suggested the numbers are changed to letters in line 32 and 33. This is also the case in 2.1 Animals selected line 98-100.

Line 34: It is suggested to change to intact females instead of entire females

Keywords:

It is suggested the keywords are alphabetic ordered.

Introduction:

In general, the clinical reasoning for when to do a TEG evaluation of a sample from a pet rabbit is missing. The manuscript describes how to perform a kaolin activated TEG analysis, but not when and in which clinical situations it could be relevant to be performed.

 Line 46+47: It is stated haemostasis prevents blood loss, which I agree, but it also prevents thrombi to be formed. It is strongly suggested this part of heamostasis (fibrinolytic part) also being mentioned.

Line 49: A reference is missing after the statement … vitro systems (Ref). When was this and in which species?

Line 51: A reference is missing after the statement …. Coagulation process (ref).

Line 58: The sentence is suggested to be rephrased. It is not the pin which is binding to the sides of the cup.

Line 60: It is suggested this phrase can be deleted.

Line 63: A verb is missing?

Line 67: Is kaolin always preferred compared to tissue factor activated TEG?

Line 69: Which hemostatic disorders have been investigated and are they similar to diseases in rabbits?

Line 71: The reference for pigs is not a primary article. It is suggested to find another reference

Line 77: Another study which could also be mentioned is Diverse coagulopathies in a rabbit model with different abdominal injuries DOI: 10.5847/wjem.j.1920-8642.2017.02.011 and The Detection of Changes in Heparin Activity in the Rabbit: A Comparison of Anti-Xa Activity, Thrombelastography®, Activated Partial Thromboplastin Time, and Activated Coagulation Time DOI: 10.1097/00000539-200212000-00008

Line 78-88: It will increase the knowledge of the reader, if it also explained which parameters will influence/be part of the explained TEG values. Eg. The R time will be dependent of the initiation of the clotting factors. The alpha angel of the crosslinking of fibrinogen and MA will depend on the platelet number and fibrinogen concentration.

Materials and methods

Line 98: It is suggested to change the word to intact females. Furthermore, was is evaluated if the two intact female rabbits were pregnant? In other animals pregnancy can influence the TEG values.

Line 108: Which body condition scorings system is used?

Line 124: It is stated the citrate tube is centrifuged. But the TEG is performed on kaolin active citrated whole blood?

Line 125: It is stated serum is used for ptt, pt and fibrinogen it assumed it is citrated plasma?

Line 127: Which analyzers for hemogram, biochemistry and coagulation parameters analyses were used and which manufacturers.

Line 135: The manufacturer of Kaolin is missing. Are the eppendorff tubes eventually pre-fabricated as the Kaolin Reagent are in 1 mL cryotubes from Haemonetics? Or how are they prepared and stored?

Line 150: How was outliers detected and defined? In table 1 it is stated it was the number of samples out of refence interval?

Also, it is also recommended the confidence interval of the lower and upper limit of reference interval are calculated to evaluate the uncertainty of the interval limits.

Results:

Line 153: The results from CBC, biochem, ptt, pt and fibrinogen are not presented. It is stated they are within normal limits. It is recommended the results are presented eventually in a supplemental table. Especially the hematocrit, platelet number and fibrinogen concentrations are important to see the results of, since they can have a large impact on the TEG results. There can be a large difference of the values in the included animals even though the values are within the reference intervals.

Line 164: LY30% is measured. How come is LY60% not measured? At line 139 you write the procedure lasts 120 minutes, so it will probably be included in most samples?

Discussion:

In general, a discussion of the values found in this study compared to other animal species are missing. Are the results in general comparable with dogs, cats or eventually pigs? And a discussion of the LY30% is recommended. A LY30 of 41,5% as described can be considered high in other animals. Also, it is recommend to discuss the outliers in general. Was it the same three animals there was out of RI for the three parameters K, alpha and LY30 or different animals?

Line 206: It has been discussed why kaolin activation can be better than native samples. But how is kaolin compared to tissue factor. It is only touched upon in cats. What in other species?

Line 230: It can also be suggested to write that no pet rabbits with bleeding/thrombotic disorder have been analyzed or it can be mentioned as a perspective.

Line 234: TEG should be as an abbreviation

Line 136: Could be deleted

Conclusion

Line 240: I partly disagree with the first line in the conclusion. I agree the study has established reference intervals for pet rabbits and the technique can be used in rabbits to evaluate the haemostasis. But it is also important to state that the study is based on only healthy animals. No animals with either bleeding or thrombosis were included, and therefore the study has not proved it will be able to detect these abnormalities. It is suggested to include eventually the word “the technique” or another phrasing.

Figure 1: Several abbreviations of parameters are visible on the figure but are not described in the figure text or in the manuscript. It is suggested only parameters evaluated in the manuscript are visible. Furthermore it can be suggested to illustrate on the figure where R, K, alpha, MA, G and LY30 are measured.

In the figure text a=angle, but no little letter a is on the figure only Angle or A

Table 1:

It can be suggested to include number of samples = n in the table.

Reviewer 4 Report

Bassan et al. present an investigation wherein a variety of domesticated rabbit strains undergo thrombelastographic analyses to determine basic baseline values of coagulation kinetics. The work is quite interesting, and determinants beyond strain include gender and age.

The authors obtained venous blood from 24 rabbits as detailed on page 3. Unfortunately, 22 of the rabbits had already been sterilized, making comparisons based on hormone changes in either gender impossible. There were also 5 different strains of rabbit assessed, but the groups were very asymmetric (e.g., n=16 in one group, with groups of n=1 in others). The differences in age and weight also varied 3 to 4-fold. In summary, the large variability by category and very small number of subjects per category renders this study without meaningful power except to document the thrombelastographic parameter ranges of pet rabbits encountered in a small veterinary practice.

The other methods described are reasonable. It should be noted that TEG assays can be crafted to take as little as 15 minutes with activators and on average 30 minutes provides all the information needed in either clinical or scientific settings.

Regarding results, the data is unambiguous and easy to understand. There can be no valid comparisons between genders, ages, etc. secondary to the lack of statistical power as noted earlier in this review. The authors need to check their data for errors – in table 1 the  average R value cannot be 315 minutes, but like is 3.15 instead. Also, please do not include any decimal places after the whole numbers – while provided by the software, it is not accurate to that degree. It should be noted that there is a fair amount of variability in the TEG results.

Where is the corresponding standard hematology data that was collected? It was described in Methods, but platelet count, fibrinogen concentration, PT, etc. are not presented. Please include them.

The authors also note in their Discussion: “Limitations of the current study were the reduced number of rabbits selected, an extended coagulation profile with thrombin time (TT), antithrombin (AT), fibrin D-dimer, and factor VIII activity.” I don’t know what this means. Again, please include all data generated by these rabbit samples.

Another issue that the authors note is that there is a paucity of previous papers presenting rabbit TEG data. The author state that there are only two. I would like to point out that there are a great many more than that in the last two decades:

1. Nielsen VG. Effects of purified human fibrinogen modified with carbon monoxide and iron on coagulation in rabbits injected with Crotalus atrox venom. J Thromb Thrombolysis. 2017 Nov;44(4):481-488.

2. Nielsen VG, Sánchez EE, Redford DT. Characterization of the Rabbit as an In Vitro and In Vivo Model to Assess the Effects of Fibrinogenolytic Activity of Snake Venom on Coagulation. Basic Clin Pharmacol Toxicol. 2018 Jan;122(1):157-164.

3. Nielsen VG. Crotalus atrox Venom Exposed to Carbon Monoxide Has Decreased Fibrinogenolytic Activity In Vivo in Rabbits. Basic Clin Pharmacol Toxicol. 2018 Jan;122(1):82-86.

4. Nielsen VG, Arkebauer MR, Wasko KA, Malayaman SN, Vosseller K. Carbon monoxide-releasing molecule-2 decreases fibrinolysis in vitro and in vivo in the rabbit. Blood Coagul Fibrinolysis. 2012 Jan;23(1):104-7.

5. Nielsen VG, Chawla N, Mangla D, Gomes SB, Arkebauer MR, Wasko KA, Sadacharam K, Vosseller K. Carbon monoxide-releasing molecule-2 enhances coagulation in rabbit plasma and decreases bleeding time in clopidogrel/aspirin-treated rabbits. Blood Coagul Fibrinolysis. 2011 Dec;22(8):756-9.

6. Nielsen VG, Huneke RB, Khan ES. Carbon monoxide releasing molecule-2 enhances coagulation in rat and rabbit plasma. Blood Coagul Fibrinolysis. 2010 Apr;21(3):298-9.

7. Nielsen VG. Hemodilution modulates the time of onset and rate of fibrinolysis in human and rabbit plasma. J Heart Lung Transplant. 2006 Nov;25(11):1344-52.

8. Nielsen VG. Effects of Hextend hemodilution on plasma coagulation kinetics in the rabbit: role of factor XIII-mediated fibrin polymer crosslinking. J Surg Res. 2006 May;132(1):17-22.

9. Nielsen VG. Effects of PentaLyte and Voluven hemodilution on plasma coagulation kinetics in the rabbit: role of thrombin-fibrinogen and factor XIII-fibrin polymer interactions. Acta Anaesthesiol Scand. 2005 Oct;49(9):1263-71.

10. Nielsen VG. Hemodilution with lactated Ringer's solution causes hypocoagulability in rabbits. Blood Coagul Fibrinolysis. 2004 Jan;15(1):55-9.

11. Nielsen VG, Crow JP. Peroxynitrite decreases rabbit tissue factor activity in vitro. Anesth Analg. 2004 Mar;98(3):668-71.

12. Nielsen VG. The detection of changes in heparin activity in the rabbit: a comparison of anti-Xa activity, thrombelastography, activated partial thromboplastin time, and activated coagulation time. Anesth Analg. 2002 Dec;95(6):1503-6.

13. Nielsen VG, Geary BT. Coagulopathy mediated by hepatoenteric ischemia-reperfusion in rabbits: role of xanthine oxidase. Transplantation. 2002 Oct 27;74(8):1181-3.

14. McCammon AT, Wright JP, Figueroa M, Nielsen VG. Hemodilution with albumin, but not Hextend, results in hypercoagulability as assessed by Thrombelastography in rabbits: role of heparin-dependent serpins and factor VIII complex. Anesth Analg. 2002 Oct;95(4):844-50.

15. Nielsen VG. Resuscitation with Hextend decreases endogenous circulating heparin activity and accelerates clot initiation after hemorrhage in the rabbit. Anesth Analg. 2001 Nov;93(5):1106-10.

16. Nielsen VG. Endogenous heparin decreases the thrombotic response to hemorrhagic shock in rabbits. J Crit Care. 2001 Jun;16(2):64-8.

17. Nielsen VG. Nitric oxide decreases coagulation protein function in rabbits as assessed by thromboelastography. Anesth Analg. 2001 Feb;92(2):320-3.

18. Nielsen VG, Armstead VE, Geary BT, Opentanova IL. Pentalyte does not decrease heparinoid release but does decrease circulating thrombotic mediator activity associated with aortic occlusion-reperfusion in rabbits. Anesth Analg. 2001 Feb;92(2):314-9.

19. Nielsen VG, Geary BT. Hepatoenteric ischemia-reperfusion increases circulating heparinoid activity in rabbits. J Crit Care. 2000 Dec;15(4):142-6.

20. Nielsen VG, Geary BT. Thoracic aorta occlusion-reperfusion decreases hemostasis as assessed by thromboelastography in rabbits. Anesth Analg. 2000 Sep;91(3):517-21.

21. Nielsen VG, Geary BT, Baird MS. Evaluation of the contribution of platelets to clot strength by thromboelastography in rabbits: the role of tissue factor and cytochalasin D. Anesth Analg. 2000 Jul;91(1):35-9.

22. Nielsen VG, Geary BT, Baird MS. Effects of DETANONOate, a nitric oxide donor, on hemostasis in rabbits: an in vitro and in vivo thrombelastographic analysis. J Crit Care. 2000 Mar;15(1):30-5.

23. Nielsen VG, Baird MS. Extreme hemodilution in rabbits: an in vitro and in vivo Thrombelastographic analysis. Anesth Analg. 2000 Mar;90(3):541-5.

The purpose of presenting these works is not so that the authors should include them all as new citations. Instead, there should be some acknowledgement of this body of work, and a contrast of type of rabbit (NZW, male (intact), 2-3 kg), activator used (celite, tissue factor, etc.) and any other relevant correlated hemostatic test included (ACT, aPTT, platelet count, etc.). Rather than include a small number of several types of rabbits, these works essentially used the same sort of rabbit.

In conclusion, the authors present a small, highly variable data set from domestic rabbits. Perhaps the authors should reconsider their manuscript as the stand-alone work presented or should perhaps should consider submitting a review article about this matter that includes some data from these studies. At a minimum, a few paragraphs that compare and contrast these works are warrented.

Round 2

Reviewer 2 Report

Thank you for the opportunity to reassess the amended version of the manuscript which in my opinion has improved considerably

Reviewer 4 Report

No further comments. Well done.

Please seek assistance from a colleague with extensive English scientific prose.